

# Genotypic variation in disease susceptibility among cultured stocks of elkhorn and staghorn corals

Margaret W. Miller[1,2], Philip J. Colburn[3], Emma Pontes[3],
Dana E. Williams[1,3], Allan J. Bright[1,3], Xaymara M. Serrano[3,4] and
Esther C. Peters[5]

[1] Southeast Fisheries Science Center, NOAA-National Marine Fisheries Service, Miami, FL, USA
[2] SECORE International, Miami, FL, USA
[3] Rosenstiel School of Marine and Atmospheric Science, University of Miami, Miami, FL, USA
[4] Atlantic Oceanographic and Meterological Laboratory, NOAA Oceanic and Atmospheric Research, Miami, FL, USA
[5] Department of Environmental Science and Policy, George Mason University, Fairfax, VA, USA

Corresponding author
Margaret W. Miller,
m.miller@secore.org

## ABSTRACT

Disease mortality has been a primary driver of population declines and the threatened status of the foundational Caribbean corals, *Acropora palmata* and *A. cervicornis*. There remain few tools to effectively manage coral disease. Substantial investment is flowing into *in situ* culture and population enhancement efforts, while disease takes a variable but sometimes high toll in restored populations. If genetic resistance to disease can be identified in these corals, it may be leveraged to improve resistance in restored populations and possibly lead to effective diagnostic tests and disease treatments. Using a standardized field protocol based on replicated direct-graft challenge assays, we quantified this important trait in cultured stocks from three field nurseries in the Florida Keys. Field tests of 12 genotypes of *A. palmata* and 31 genotypes of *A. cervicornis* revealed significant genotypic variation in disease susceptibility of both species measured both as risk of transmission (percent of exposed fragments that displayed tissue loss) and as the rate of tissue loss (cm$^2$ d$^{-1}$) in fragments with elicited lesions. These assay results provide a measure of relative disease resistance that can be incorporated, along with consideration of other important traits such as growth and reproductive success, into restoration strategies to yield more resilient populations.

## INTRODUCTION

Disease constitutes an existential threat to coral persistence (*Walton, Hayes & Gilliam, 2018*), and this threat is exacerbated by its worsening with thermal stress (*Randall & Van Woesik, 2015*). Studies correlating various life history traits with disease susceptibility have shown that the family Acroporidae, with its high skeletal growth rates and low investment in immunity, is particularly susceptible to disease (*Palmer, Bythell & Willis, 2010*; *Díaz & Madin, 2011*). The two species of *Acropora* native to the Caribbean,

*A. palmata* and *A. cervicornis*, are both designated as critically endangered by the IUCN and threatened under the US Endangered Species Act (ESA) with disease cited as a primary driver of their high extinction risk (*Aronson & Precht, 2001*; *NMFS, 2006*). The strong association of climate change related thermal stress and disease is also well-documented in both these species (*Muller et al., 2008*; *Randall & Van Woesik, 2015*), lending little expectation of abatement under expected continued warming.

Legal mandates imposed by the ESA listing as well as recognition of persistent patterns of population decline have yielded increasing investment and success in population enhancement for *A. cervicornis* and, more recently, *A. palmata* (*Lirman & Schopmeyer, 2016*; *Miller, Kerr & Williams, 2016*). Currently, population enhancement efforts focus on fragment-based (clonal) propagation methods, yielding relatively low genotypic diversity (e.g., up to 100 genotypes utilized in a regional population enhancement program) and potentially higher disease susceptibility (*Altizer, Harvell & Friedle, 2003*). However, wild remnant populations for these species often display even lower genotypic diversity (e.g., individual reefs occupied by a single genotype; *Baums, Miller & Hellberg, 2006*). Disease takes a variable but sometimes high toll on restored populations, at a level comparable to that on wild remnant populations (*Miller et al., 2014*). The current lack of tools for management or mitigation of disease impacts in either wild or restored populations underscores the potential benefits to identifying natural disease resistance within populations. Indeed, *Vollmer & Kline (2008)* identified 6% of *A. cervicornis* genotypes as being resistant to disease in a wild population in Panama based on field surveys and field-based challenge assays and, more recently, *Muller, Bartels & Baums (2018)* identified 27% of *A. cervicornis* genotypes in nursery culture in the lower Florida Keys as disease resistant in laboratory challenge assays using a homogenized inoculant. No published studies have tested such resistance in *A. palmata*.

Identifying genotypes with specific disease susceptibility or resistance traits could provide an important tool in the quest to build more resilient, recovered populations of these threatened species. Also, the quantification of these disease susceptibility traits among genotypes is a pre-requisite to further investigation of underlying biological and/or genomic mechanisms (*Libro & Vollmer, 2016*) that may eventually lead to effective diagnostic tests and disease treatments. Thus, we performed field challenge assays to quantify disease susceptibility or resistance in a range of genotypes from stocks cultured in three upper Florida Keys coral nurseries.

## METHODS

### Susceptibility assays

We performed field challenge assays according to a protocol that was previously illustrated and published to facilitate standard trait quantification by other researchers or nursery operators (*Miller & Williams, 2016*). All assays were performed on segregated, experimental coral nursery "trees" (*Nedimyer, Gaines & Roach, 2011*) at the nursery operated by the Coral Restoration Foundation off Tavernier, Florida, USA. All nursery stocks had been previously genotyped (via microsatellites or direct sequencing of mitochondrial or nuclear genes; see Tables S1 and S2) and tracked through propagation

via best practices (*Johnson et al., 2011*). Briefly, replicate fragments of the tested genotypes were collected from the nursery population and deployed to the experimental "trees" in a segregated area of the nursery. After at least two weeks' stabilization period, actively diseased fragments were collected from the nursery population ("inoculants"; all *A. cervicornis*) and attached securely to the apparently healthy test fragment with cable ties. The genotype of inoculant fragments was not recorded or controlled, though multiple genotypes of inoculant were used in each trial. However, we applied all inoculants haphazardly across replicates of different genotypes to minimize potential bias; within each trial, the sequential collection of inoculant fragments (which was clumped by the nursery tree of origin and, hence, genotype) was distributed across replicate test fragments of different genotypes.

Hereafter, we use the term "transmission" to refer to the appearance of tissue loss signs on a test fragment after the application of an inoculant, though the use of this term does not imply anything about the specific mechanism or pathogen that might have caused the tissue loss. Bacteria, ciliates, and suspect viruses have all been detected in cases of Caribbean *Acropora* tissue-loss disease (*Miller et al., 2014*; *Sweet, Croquer & Bythell, 2014*; E. C. Peters, 2015, personal observation). However, the responsible biotic pathogen may shift over time (*Sutherland et al., 2016*) and/or a dysbiosis (i.e., alteration of normal microbiome) or a noninfectious abiotic factor (e.g., elevated or lowered sea temperatures or chemical contaminant) (*Lesser et al., 2007*) may be the dominant agents of disease in *A. palmata* and *A. cervicornis*.

The assays were intended to be surveyed (examined for occurrence of tissue loss in the test fragments, photographed, and elicited lesions measured) on day 1, 3, and 5 after implementation (day 0) and the trial was completed on day 7, when inoculants were removed and discarded, along with all lesioned fragments. Slight deviation in timing of surveys was dictated by extraneous events (weather or other disturbance). In accordance with permit conditions, test fragments that did not show signs of tissue loss were either discarded or remained quarantine for several more weeks and then transferred back to nursery stocks or laboratory studies.

A total of 16 *A. cervicornis* and six *A. palmata* genotypes were screened in two 1-week trials between July and August 2016, with an additional two trials run in July–August 2017 (additional 14 *A. cervicornis* and six *A. palmata* genotypes). Three response parameters were analyzed and compared among genotypes in the transmission assays for each year and species separately. Risk of transmission (7 $d^{-1}$) was expressed as the proportion of assay replicates in which tissue-loss lesions were observed ($n = 7$–$10$ for genotypes tested in 2016, $n = 10$ for all 2017 genotypes). This proportion for each genotype was compared, via Pearson Chi-squared tests, to an "expected" value based on the pooled population of all the genotypes assayed in the same year. Second, the time to first appearance of tissue loss (i.e., survey day on which lesion was first observed) was averaged for transmitted replicates of each genotype with at least three lesioned replicates and compared via Kruskall–Wallis ANOVA. Third, the length of all elicited lesions was measured with a ruler *in situ* at each survey. The ending length of each lesion (i.e., at day 7) was divided by the number of days since the lesion was first observed. These rates

(i.e., progression of tissue-loss margin; cm d$^{-1}$) were averaged for each genotype with at least three lesioned replicates and compared among genotypes via a Kruskal–Wallis ANOVA. To obtain higher replication for the latter two parameters in 2016 (when overall transmission rates were lower than 2017), fragments that did not experience tissue loss in the initial trial were re-exposed to a second inoculant trial. Replicates that showed elicited lesions in this repeat assay were included in the reported tissue-loss rate and time to transmission data but were not included in the risk of transmission score (since they had experienced added inoculation "dose").

## Inoculant characterization and controls

*Acropora cervicornis* fragments were used as inoculants in the assays for both species because no disease was observed in the background nursery *A. palmata* population. Since there are no definitive field diagnostic tools for *A. cervicornis* disease identification, we performed basic histopathological observations to characterize the disease inoculants used in our assays. On day 0 of each transmission trial, additional diseased fragments (comparable to those used as inoculants; $n = 13$ *A. cervicornis* in 2016; $n = 3$ in 2017) were fixed in a solution of one part zinc-buffered formalin (Z-Fix; Anatech, Ltd., Battle Creek, MI, USA) diluted with four parts seawater. Samples were shipped to the Histology Laboratory at George Mason University for processing and slide reading according to the protocols described in Miller et al. (2014). Additional histoslides cut from one representative sample from each year were treated with the eubacterial probes EUB338-I and EUB338-II, tagged with AlexaFluor 488 and CY3 fluorescent dyes (Creative Bioarray, Shirley, NY, USA), respectively, and a nonsense probe (nonEUB338) to investigate suspect bacteria. After applying the probes to separate deparaffinized and hydrated slides in a 45% formamide hybridization buffer solution, the samples were heated to 55 °C for 2 h, rinsed with wash buffer, and coverslipped with FluoroshieldTM with DAPI aqueous mounting medium (ImmunoBioscience Corp., Mukilteo, WA, USA) before examining with a Leica DM2000 fluorescence microscope and integrated camera.

The progression of tissue loss on a given diseased branch is often intermittent in *A. cervicornis*, with unknown effect on its potential infectivity. We chose the visually most active tissue-loss margins we could find (i.e., widest margins of bright white exposed skeleton and ragged or "sloughing" appearance of the tissue margin) as inoculants. We applied all inoculants haphazardly across replicates of different genotypes to minimize potential bias from differentially infective inoculants. In 2017 we also used photographs and measured lesion size at the beginning and end of each trial to verify the progression of tissue loss on the inoculant fragments.

Note that the purpose of this study was to quantify tissue-loss disease susceptibility as a phenotypic trait of specific genotypes, not to characterize the potential pathogen nor transmission mechanism per se. Hence, a full slate of "healthy inoculant" control treatments was not undertaken. Nonetheless, a small number of "control" assays was conducted in 2016 using healthy-looking *A. cervicornis* "inoculants" paired with one fragment each of four test genotypes of each species. These *A. palmata* controls ($n = 4$) represent healthy allografts (i.e., healthy-looking *A. cervicornis* "inoculants" applied to

an *A. palmata* test fragment) to detect potential allo-rejection response independent of disease exposure.

Field experiments and sample collection were performed under Florida Keys National Marine Sanctuary permit #FKNMS-2016-024-A1.

## RESULTS

None of the "healthy control" assays conducted in 2016, including the allograft controls, resulted in development of tissue-loss lesions or any visually observable anomalies on the test fragments, though minor abrasion on protruding calices was sometimes observed (Figs. S1A and S1B).

For the 2017 trials, we noted progression of the tissue-loss margin of each inoculant and 81% of them lost noticeable tissue during the 7-day trial ("active" inoculants). If the other 19% of replicates involving inoculants without observable tissue loss ("inactive" inoculants) are excluded, the range of transmission risk among tested genotypes was 38–100%, compared to 30–100% when all replicates were included. Also, we observed multiple instances (5 out of 27) in which an inactive inoculant was associated with transmission, suggesting that lack of progressing tissue loss did not preclude its ability to cause tissue loss (i.e., serving as an effective inoculant). For these reasons, we included all replicates in results reported here. We observed greater virulence of the inoculant condition, as manifested by a higher overall risk of transmission, in the 2017 trials (56% transmission for *A. cervicornis*, 82% transmission for *A. palmata*) than in 2016 (30% for each species).

As expected, considerable variability was observed in the characterization of the inoculant samples submitted for histological examination. A total of 11 of these 13 inoculant samples from 2016 were characterized as white-band disease (WBD) from their gross signs (i.e., the disappearance of coral tissue along a smooth margin, mostly starting from the base of the branch, although two of these had tip lesions that may have been caused by predation). These fragments had diverse microorganisms trapped in the protective agarose layer that was applied over the tissue-loss margins prior to decalcification and processing, including *Symbiodinium*-containing ciliates adjacent to sites where gastrodermal cells had been stripped from the mesoglea, then the epidermal cells lysed or sloughed off (Fig. S2A). However, other areas lacked ciliates. Intracellular Rickettsiales-like organisms (RLOs) occurred in mucocytes of polyp structures (Fig. S2B), but only one of these (grossly) WBD samples had excess mucus production in epidermal mucocytes. Tissues also varied in the formation of single-cell necrosis or apoptosis and degraded cell spherules at the tissue-loss margins. The remaining two samples from 2016 and all three sampled from 2017 were grossly characterized as rapid tissue loss (RTL), having acute patchy sloughing of tissues off the branches; denuded skeleton was bright white, lacking fouling organisms. These RTL samples were in poorer condition, with more epidermal mucus production (Fig. S2C) and suspect bacteria (based on Giemsa staining) in the mucus (Fig. S2D), some ciliates, and lysing and necrosis of atrophied epithelia. Along the tissue-loss margin, portions of the calicodermis consisted of thin columnar, hypertrophied cells with abundant eosinophilic apical granules of coral

acid-rich proteins (Fig. S2E). The tissues displayed larger foci of single-cell necrosis or apoptosis in basal and surface body walls (Fig. S2F) and formed necrotic cell spherules at the tissue-loss margin (Fig. S2G). Fewer ciliates were associated with the 2017 samples. However, as has been found in Caribbean acroporid samples affected by either type of tissue-loss pattern (E. C. Peters, 2012, personal observation), sections through tentacles revealed abundant zooxanthellae in the gastrodermis (as expected), but the gastrodermal cells were also variably filled with minute turquoise-staining (with Giemsa) specks, in some places being released through the apical surfaces of the gastrodermal cells into the gastrovascular cavity fluid. The gastrodermis areas in the tentacles with the densest aggregations of these specks had fewer algal cells (Fig. S2H).

The FISH procedure showed positive staining with the bacterial probes by abundant minute coccoid or other-shaped cells on the surfaces of necrotic or apoptotic areas (Figs. S3A–S3D). RLOs were present in the mucocytes of the tentacles, oral disc, actinopharynx epidermis, and cnidoglandular bands, with similar densities of infected cells in 2017 samples as found in all of the 2016 samples. The FISH probes revealed differences in the distribution of bacteria between the samples, however, as well as morphologies of the bacteria (Figs. S3A–S3D). The most degraded tissues along the tissue-loss margins had abundant minute bacteria that were labeled by both the EUB338-I and EUB338-II probes (Figs. S3E and S3F).

Despite variable inoculant virulence, significant variation was observed among genotypes of each species in risk of transmission (Figs. 1A and 1B). In 2016, four of 16 *A. cervicornis* genotypes and one of six *A. palmata* genotypes had significantly higher rates of transmission than expected (Pearson Chi-squared test, $p < 0.05$; Fig. 1A). Two additional *A. cervicornis* genotypes and one additional *A. palmata* genotype appeared relatively resistant to disease (i.e., zero tissue loss in $n = 7–10$ replicate exposures) in 2016, although the power of the Pearson Chi-squared tests was not adequate to distinguish these from the pooled population transmission risk of 30% with 95% confidence (Fig. 1A). In 2017, three of 14 tested *A. cervicornis* genotypes had significantly higher risk of transmission than the population expectation of 56%, whereas none of the six tested *A. palmata* genotypes differed significantly from the 82% expected transmission risk (Fig. 1B).

Mean tissue-loss rates for elicited *A. cervicornis* lesions ranged from ∼0.5 to 4 cm d$^{-1}$ in 2016 and from ∼1 to 7.6 cm d$^{-1}$ in 2017-tested genotypes (Figs. 1C and 1D). This represented statistically significant variation in 2016 ($p = 0.022$, Kruskal Wallis ANOVA) but not in 2017 ($p = 0.135$). Tissue-loss rates in *A. palmata* did not show significant variation among genotypes in either year (Figs. 1C and 1D). Similarly, "days to transmission" did not differ significantly among genotypes for either species tested in either 2016 or 2017 (Figs. 1E and 1F).

## DISCUSSION

The prior work on which this study was based had shown 6% of 49 wild *A. cervicornis* genotypes in Panama to be "resistant" to what the authors referred to as WBD (*Vollmer & Kline, 2008*). This designation of resistance was based on ∼quarterly surveillance of

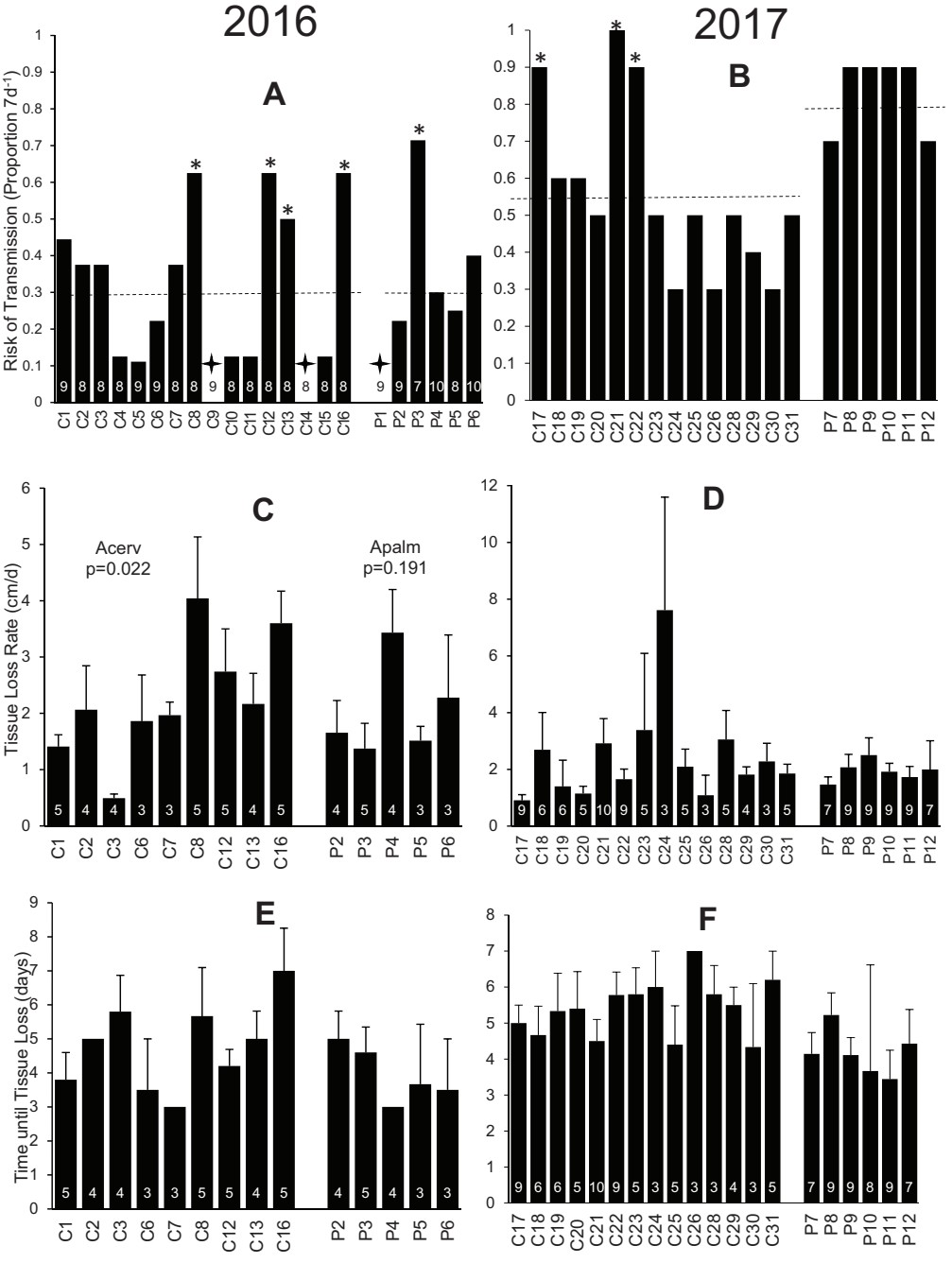

**Figure 1 Results of disease susceptibility assays conducted in 2016 (A, C, E) and 2017 (B, D, F).** (A and B) the risk of transmission (i.e., the proportion of replicate ramets of each genotype in which the elicitation of a tissue loss lesion was observed following the application of an inoculant fragment); (C and D) the average rate of progression of tissue loss for genotypes with at least three responding replicates; and (E and F) the average time until a lesion was observed for genotypes with at least three responding replicates. Genotype designations (given along the *x*-axis) are identified in Table S1. The number of replicates for each genotype is given inside or above each bar while error bars indicate +1SE. In A and B, dashed lines show the pooled population risk of infection for each species, asterisks indicate genotypes that differed significantly from this expected transmission risk, and crosses indicate zero transmission observed for that replicate (though power for the Pearson Chi-squared test was inadequate to conclude that zero differed from expected).

disease prevalence in this wild population over a 1-year period and a score of zero transmission during a 3–5-day challenge assay with replication of five. A more recent study (*Muller, Bartels & Baums, 2018*) using laboratory challenge with a homogenized tissue inoculant (and replication of five to seven) determined 27% of *A. cervicornis* genotypes (four of 15) from a nearby coral nursery (lower Florida Keys) were resistant under background conditions, although most lost resistance during acute heat stress/bleaching. Only one of these four genotypes remained disease-resistant when bleached by a warm thermal anomaly affecting this nursery in September 2015 (*Muller, Bartels & Baums, 2018*).

Our results in 2016 suggested that 12.5% of the assayed *A. cervicornis* and 17% of *A. palmata* genotypes were resistant (i.e., zero transmission of disease signs in $n = 8$ or 9 replicate exposures). However, none of the screened genotypes showed zero transmission in 2017. This shift to less observed resistance in the second year is unlikely to be due to differential thermal stress/bleaching as no substantial difference in thermal stress (i.e., overall average at nearby Molasses Reef between 1 July and 20 August was 29.8°C for 2016, 29.9 for 2017; https://www.ndbc.noaa.gov/station_page.php?station=mlrf1) and no bleaching was observed in the nursery populations during either year. Instead, we suggest that this pattern resulted from either including more susceptible test genotypes in 2017 and/or the presence of more virulent disease condition(s) in the background population and hence the inoculants used in 2017. Something in those inoculants caused thick mucus discharge (whether bacterial or toxin-mediated) and was acutely degrading the tissues to rapidly form large areas of single-cell necrotic or apoptotic changes. The differences revealed by FISH suggest that the microbial communities were potentially different, not only in the quantity of bacteria present, but also in species present, as revealed by their morphologies. Molecular methods are needed to understand these changes, but may be difficult and expensive to apply during assays.

This emphasizes the likelihood that field-collected inoculants, though visually similar and collected from the same location/population, may not represent the same condition. Our assays relied on visual determination of active disease signs to identify effective inoculants (due to the lack of available reliable field diagnostic tools for *Acropora* diseases) and we did not control nor track the genotypes of the inoculants. Nonetheless, the lack of allogenic response in the healthy controls and the haphazard pairing of inoculants across test genotypes with relatively high replication make it quite unlikely that the statistically significant variability in disease susceptibility observed within each assay trial was spuriously caused by the uncontrolled genotype of the inoculant. It does, however, suggest that results of this protocol should be interpreted as indicating *relative* disease susceptibility, rather than absolute resistance, and confidence in this relative susceptibility ranking is highly dependent on replication. Additional histological and molecular microbiological studies of coral fragments used in repeat susceptibility assays under different conditions would also improve the interpretation of "relative resistance."

Contrasting tissue-loss rates have been cited in past literature as potentially diagnostic signs for particular coral diseases (*Richardson et al., 1998*). However, the current results indicate that tissue-loss rate can be characteristic to particular genotypes within the same disease event. Interestingly, a few *A. cervicornis* genotypes showed tissue-loss rates

contrasting with their risk of disease signs transmission, suggesting the possibility of multiple mechanisms of resilience to disease effects. For example, genotypes C3, C17, and C22 showed higher than expected risk of transmission, but relatively low tissue-loss rates, perhaps indicative of a strategy of disease (tissue loss) tolerance rather than resistance.

Disease resistance, along with other beneficial traits, is commonly discussed as a tool to leverage greater resilience and prospects of long-term recovery in restored coral populations. The characterization of a range of fitness traits (growth, reproduction, as well as disease or thermal resistance) is important to consider in selecting genotypes for propagation as ecological tradeoffs or unanticipated environmental stressors may yet compromise the success of disease-resistant individuals. Field studies include cases where disease resistance is associated with both positive (e.g., high growth rates) and negative (reduced thermal tolerance) traits (*Shore-Maggio, Callahan & Aeby, 2018*). It may be appropriate to preferentially include a disease-resistant genotype in an outplanting design (e.g., inclusion across all sites), but it must be incorporated in diverse patches. Diverse provenance of outplanted coral populations gives the greatest likelihood of successful fertilization and larval production as well as greatest adaptability of these offspring. This quantification of variation in disease-resistance can also serve as a basis for mechanistic studies, potentially leading to much-needed effective disease management tools.

## CONCLUSION

This study has documented genotypic variation in disease susceptibility, as evidenced by risk of transmission and tissue loss rates, in both Caribbean acroporid species that are highly targeted in population enhancement efforts. This provides a context for continuing genomic investigations of mechanism and potential development of diagnostics (Traylor–Knowles and Young, in progress). In combination with quantification of other phenotypic traits, this result also provides the potential to leverage this trait in restored populations to increase resilience, but in a strategy that maintains genetic diversity.

## ACKNOWLEDGEMENTS

The collaboration and support of the three regional coral nurseries operated by the Coral Restoration Foundation, the Florida Fish and Wildlife Research Institute, and the University of Miami are gratefully acknowledged. Field assistance from R. Pausch, and A. Peterson is also greatly appreciated. K.A. Cobleigh, E.V. Mazur, and V.T. Nguyen prepared the tissue samples for histopathological examination.

### Funding

This project was supported by the National Oceanic and Atmospheric Administration's Coral Reef Conservation Program and NOAA's Southeast Fisheries Science Center.

The funders had no role in study design, data collection and analysis, decision to publish, or preparation of the manuscript.

### Grant Disclosure
The following grant information was disclosed by the authors:
National Oceanic and Atmospheric Administration's Coral Reef Conservation Program and NOAA's Southeast Fisheries Science Center.

### Competing Interests
The authors declare that they have no competing interests.

### Author Contributions
- Margaret W. Miller conceived and designed the experiments, performed the experiments, analyzed the data, prepared figures and/or tables, authored or reviewed drafts of the paper, approved the final draft.
- Philip J. Colburn performed the experiments, analyzed the data.
- Emma Pontes performed the experiments, analyzed the data, prepared figures and/or tables.
- Dana E. Williams conceived and designed the experiments, performed the experiments, approved the final draft.
- Allan J. Bright performed the experiments, approved the final draft.
- Xaymara M. Serrano conceived and designed the experiments, approved the final draft.
- Esther C. Peters analyzed the data, contributed reagents/materials/analysis tools, prepared figures and/or tables, authored or reviewed drafts of the paper, approved the final draft.

### Field Study Permissions
The following information was supplied relating to field study approvals (i.e., approving body and any reference numbers):
Field experiments and sample collection were performed under Florida Keys National Marine Sanctuary permit #FKNMS-2016-024-A1.

### Data Availability
The raw data, constituting observations of lesion initiation and size over time in assay replicates, is available in File S1.

### Supplemental Information
Supplemental information for this article can be found online at http://dx.doi.org/10.7717/peerj.6751#supplemental-information.

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
