# Peer review of "Genotypic variation in disease susceptibility among cultured stocks of elkhorn and staghorn corals"

_PeerJ, doi:10.7717/peerj.6751_

## Round 0.1 · original submission · Minor Revisions

I now have reviews back from two expert referees and as you will see, one was more critical but both are supportive of your work being publishable following revision. Each comment that they would like to see the genotype of the inoculant reported, and in particular the first referee is concerned that the results used to identify ‘susceptible’ and ‘resistant’ genotypes may be confounded by the inoculant method used, and that lower or higher susceptibility are actually the result of differentially infective inoculants. I can see their point, and imagine other readers many have the same concern, thus I particularly encourage you to address this concern in your revision of the manuscript.

Beyond that particular concern, which seems to me the basis of the recommendation for a major revision, the suggestions of each reviewer are relatively straightforward. As such, I am returning a decision of minor revision, with the expectation that you will be able to address this major concern in a revision.

Reviewer 1 ·

Basic reporting

This manuscript describes the results of direct-graft challenge assays with multiple genotypes of Caribbean acroporid corals in Florida, to determine whether genetic resistance to disease can be identified. This manuscript is very well written in professional English, is clear, concise, and thorough. It provides clear justification for each experimental decision, and with a few exceptions outlined below, provides ample methodological information to be repeated in the future. It is self-contained and follows a professional structure, including providing sufficient background, context and references to the literature.

This work adds to the growing body of literature around identifying ideal candidates for coral restoration programs and adds to the body of basic literature around disease ecology. It also uses standardized field protocols, which advances the field by using comparable methods across studies.

Experimental design

Overall, the experimental design is sound, sufficiently replicated and appropriately analyzed. The major decisions in the design process are adequately justified, and the paper reports original primary research with clear aims and scope. The only significant concern I have regarding the design is in the use of inoculants. While the authors address the use of inoculants directly in the text (i.e. lines 144, 148 and others), I still think there is the potential for a significant bias from differentially infective inoculants.

I understand why inoculants were collected without regard to genotype, and that it is extremely difficult to know the potential infectivity of an inoculant in the field, but was there any attempt to use inoculants from ‘donor’ colonies evenly across receiving genotypes? How many different inoculant ‘donor’ colonies were used, and was there any pattern in the receiving corals related to the inoculant ‘donor’ colonies? At the very least, I think it would be useful to provide the data for which inoculant ‘donor’ colonies were used in challenge assays for each genotype, and whether there appeared to be a donor effect. Perhaps the results could be standardized in some way (maybe by rate of tissue loss on the inoculant for 2017)? In line 203, the authors write “despite variable inoculant virulence, significant variation was observed among genotypes of each species in risk of transmission’. However, if the results were not standardized to inoculant virulence, I’m not convinced that the variation in infectivity can be attributed to the genotypes of the infected corals. If they could be standardized in 2017 by rate of tissue loss on the inoculant, and the results are the same as the pre-standardization, it would add justification for this method.

This is particularly concerning given that some fragments that did not experience tissue loss in the initial trial in 2016 were re-exposed to a second inoculant. This would suggest that perhaps the authors were not convinced that the first inoculant was infective. Furthermore, only 81% of inoculants appeared to lose tissue during the 7-day trial, although I do acknowledge that some inactive inoculants were still associated with transmission (very interesting!), and appreciate the authors providing the range of transmission risk including and excluding these inactive inoculants. Thirdly, the histology of the samples suggested potentially two different diseases were impacting the inoculants (WBD and RTL), which could also result in differential transmission or level of infectivity. Therefore, I think a further justification of why these results are reliable despite potentially differentially infective inoculants is necessary.

Validity of the findings

Overall, I think the sample sizes are robust, the statistical methods are sound and there were proper controls. However, I do question whether the results are confounded by differentially infective inoculants, as discussed in the section above. I think a further justification for the use of these inoculants, or additional information about how many and which inoculant donor colonies were used for each genotype, is necessary.

Additional comments

Introduction
• Line 62, “et al” missing appropriate punctuation
• Line 56, it may be worth mentioning that restored populations also often involve the asexual fragmentation of corals that can potentially result in populations with lower genetic diversity than natural ones, and consequently higher risk of epidemics (Altizer et al. 2003 TREE 18(11), pp.589-596), although I do recognize that natural Acroporid populations, especially in Florida, currently have very low abundance and genetic diversity.
• Line 64 and Line 70 – in one instance “susceptibility/resistance” is used, and in the other instance “susceptibility or resistance” is used. I suggest keeping text consistent and prefer the latter.

Results
• Line 177, acronym WBD was used here but I don’t think it was previously defined in the text. Please define white band disease.
• Line 179, I suggest changing ‘contributed by’ to ‘caused by’
• Line 183-185, I think this may be an incomplete sentence. The first phrase seems unfinished.
• Line 187, acronym RTL was used here but I don’t think it was previously defined in the text. Please define rapid tissue loss.
• Line 196, is ‘intensities of infections’ an epidemiological term? I think it may be useful to define infection intensity in this context.

Figures & Tables
• Figure 1, it was not clear to me what the p-values in panel C are referring to.

Reviewer 2 ·

Basic reporting

no comment

Experimental design

no comment

Validity of the findings

no comment

Additional comments

This is a very nice study examining general tissue loss disease resistance in A. cervicornis and A. palmata using established techniques. Overall, the paper is well written, methods are appropriate and conclusions well supported. I commend the authors for including histology to examine the inoculate fragments but would like to see representative pictures showing the different host responses as interpreted from the slides e.g. RLOs, lysed cells, apoptosis, degraded cell spherules, etc. It could be a supplemental Figure but would really help in the evaluation of the findings and more importantly be a tool to help train others in histo slide interpretation.

I would also like them to include the genotypes of the inoculant fragments to Table 2. There may be some interactions between different genotypes that are affecting susceptibility in this experimental design and that added information would allow folks to evaluate that.

They state that the higher transmission rates in 2017 may be due to more infective inoculants which may be but it could also just be more susceptible genotypes were tested in 2017….or both.

I also appreciate them bringing the concepts of genotype diversity into restoration activities as many studies to date have only focused on a few traits e.g. bleaching resistance without considering disease susceptibility. Hopefully, this paper will bring this problem out in the open so others can learn from it.

---

## Round 0.2 · accepted · Accept

Thanks for your revised submission. I have read the manuscript and am satisfied with how you have reported your approach in the revised manuscript. Given that you did not record the genotype of the innoculant, how you report it in the revised manuscript seems the best compromise to the concerns of the referee and I agree with your approach. Given the clear explanation of your collection and application of innoculants in the body of the revised manuscript, I feel readers can draw their own conclusions, and this is ready to move forward into production.

#